# Danube River Cruises as a Strategy for Representing Historical Heritage and Developing Cultural Tourism in Serbia

**Nataša Danilović Hristić \***, **Nebojša Stefanović** and **Saša Milijić**

Institute of Architecture and Urban & Spatial Planning of Serbia, 11000 Belgrade, Serbia;
nebojsa@iaus.ac.rs (N.S.); sasam@iaus.ac.rs (S.M.)
**\*** Correspondence: ndanilovichristic@gmail.com

**Abstract:** Podunavlje, a region located along the Danube River in Serbia, features a very rich cultural heritage with different notable periods ranging from prehistoric to medieval times. It is also a unique and valuable region with natural beauty. Our research concentrates on this area, using a case study as the methodology. The starting hypothesis is that strategic orientation to develop tourist docks for river cruises and marinas along the corridor of 588 km, at certain locations, primarily in urban centers and near the prominent archaeological sites, will form conditions for the better accessibility and presentation of priceless cultural treasures. By collecting data about various trends in the region and comparing these data with international studies, the authors support the utilization of an integrated method derived from the context of sustainable spatial planning. Such a method would support the creation of suitable conditions for the reconstruction, presentation, and development of creative tourist offerings. The urban plans that regulate these areas must observe and harmonize all aspects, especially the conditions for preserving heritage alongside the need for creating new accompanying content and events that will stimulate the economy and thereby ensure self-preservation and protection. The goal of the strategic analysis presented here is to determine which tourist and cultural offers are effective and thus should be promoted. The purpose of this study is to indicate the steps and required conditions for the implementation of suitable strategies. Apart from decision making on the strategic level and the implementation process, it is necessary to consider further impacts and investigate other possibilities to fully utilize the potential in the region.

**Keywords:** river cruises; historical heritage; spatial and urban planning; strategy; tourism; development

## 1. Introduction

### 1.1. Theoretical Framework for River Cruise Tourism

When cruise tourism is mentioned, even in academic papers, it is usually about sea and ocean cruising destinations (commonly the Mediterranean, northern European fjords, Caribbean, Pacific, and other areas) and the impact of mass tourism on local attractions. River (and canal) cruise tourism seems to be neglected in the literature, or at least limited to rare locations and topics. This is surprising considering that cruise tourism was first studied a long time ago and is a popular topic, especially in Europe, with rivers like the Rhine, Danube, Loire, and Volga. The differences between sea and river cruises are evident. In a technical sense, routes vary between linear and circular routes, the sizes of ships and numbers of passengers and crew onboard are different, the durations of travel and numbers of stops are different, and the port demands are also different. There are also differences in terms of passenger preferences and experiences. Supporting this, Tomej and Lund-Durlacher observed that "not

only is the ratio between the size of a river vessel and the body of water much smaller when compared to ocean cruises, but the dependence on the resources of natural and built environments of waterways is also more prominent. While gigantic ocean liners are becoming attractions or even destinations of their own, river landscapes, with their aesthetic beauty and resources found in riverside towns and cities, remain the primary destinations attracting river cruise tourists" [1] (p. 1). Erfurt-Cooper gives the explanation that the "scholarly literature does not provide a formal definition for river cruises, but it is clear that the term refers to multi-day voyages for leisure purposes on navigable inland waterways, such as rivers and canals, sometimes including lakes, where the passengers spend the night on board, excluding river day cruises and river excursions" [2] (pp. 95–117). According to Van Balen, Dooms, and Haezendonck, "river cruises are multi-day journeys to appealing riverside destinations and ships typically offer onboard suites, dining facilities, and leisure amenities" (pp. 71–79).

The development of passenger river transport in Europe started in the 19th century with steamboats within the Habsburg Empire and even internationally, establishing a tradition that grew to become river cruising. In Europe, "floating roads" with a total length greater than 37,000 km connect hundreds of cities and industrial regions. Van Balen et al. provided data to support the claim that "the industry experienced a 10% annual growth between 2007 and 2012, with the greatest growth occurring with western European waterways, but the popularity is not limited to only this region. For ports in the proximity of popular destinations, opportunities exist as well. Additionally, river cruise operators are actively developing new journeys in order to retain old customers and attract new customers. For this reason, non-traditional river cruise ports are increasingly visited, including ports in cities that are not located along popular itineraries" [3] (pp. 71–79). Tomej et al. provided other useful data via the Cruise Lines International Association 2019 report, where, in 2018, the European region accounted for almost 90% of global river cruise passengers, of which 64% belonged to the Danube, the Rhine, and their tributaries. The European Union also holds 41% of all river cruise vessels (Central Commission for the Navigation of the Rhine, 2019 (p. 2). The economic impacts, significance, and benefits of cruising tourism have been researched by a number of authors, including Dwyer [4,5]. The topic of urban reconstruction and regeneration, especially in large projects for waterfronts, was researched by McCarthy [6]. Others, like Fachrudin and Lubis, refer more to social issues and influences on inhabitants [7].

Two other important topics related to river cruises on a wider level are sustainability and environmental impacts in general [8,9]. Connectivity to cultural landmarks along rivers, such as historical towns, castles, monasteries, or wine regions has been studied by Steinbach [10]. We may consider water traffic as one of the oldest economical and potentially environmentally sustainable ways of transporting passengers and cargo. River transport plays a vital role in economic development. In addition, proponents of water traffic claim that is the environmentally cleanest and safest transport method, with numerous development opportunities, although there are issues connected with cruising industry regarding mass usage and pollution control. Cultural tourism as a topic is well represented in academic papers by numerous authors, where it has become a vital part of location branding and development. For example, Richards discussed the definition and origin based upon the need to visit and experience the most important cultural locations for education and pleasure, and later the motivations, attractions, and frequency of visits [11–13]. Chen and Rahman analyzed engagement, contacts, memorable moments, and destination loyalty [14]. Binkhorst and Den Dekker provided an innovative perspective on tourism based on the principle of co-creation and innovation in different project settings [15]. Jovicic offered an explanation that cultural tourism "in the initial stage of its development, it represents one of the alternative forms of tourism opposed to mass tourism. The beginning of the 1990s indicates a period of transformation for cultural tourism which, unlike the original orientation towards elite clientele, found a new opportunity for development in the orientation towards the mass market because the creation and marketing of cultural attractions became a development option for numerous destinations. Cultural tourists demonstrate a proactive approach to meeting their needs, wanting to actively participate in creating experiences while travelling" [16]

(pp. 605–612). This claim is important because, as Jones et al. concluded, "river cruises are a limited onboard offer, meaning passengers spend more time on shore, predominantly participating in organized excursions included in the cruise package or offered as optional components" [17] (pp. 61–71).

### 1.2. Methodological Framework and Technical Research Details

The academic relevance of our study is that it contributes to understanding Danube cruise tourism, which is a special hybrid in terms of its cultural, natural, and urban locations. In this study, the focus of the research is on an Eastern European area, i.e., Serbia. This study considers a wide methodological frame, using a multidisciplinary approach by combining the knowledge fields of tourism, the protection of public goods, the preservation and presentation of cultural heritage, and spatial and urban planning. This integrated method lends itself to the context of sustainability, featuring the assessment of benefits and disadvantages, as well as observing and harmonizing spatial demands. Such a method is very commonly used in spatial planning [18]. Stefanović et al. argued that methodologies for spatial planning for tourist products in protected areas should contain "the following planning criteria for commercial tourism: Achieving a high standard for the tourism offer in the area, while at the same time presenting the protected resources, organizing the activities and development of the area for specific forms of tourism, achieving a dispersed distribution for the tourist reception or accommodation and protection regime, and transport and functional connections for the tourism offers in tourism zones". Methodologies for special area planning purposes have "guiding principles of sustainable development, founded on regionally balanced development goals. These goals include the promotion of territorial and social cohesion through steady social and economic development and competitiveness, improvements in traffic communication and accessibility, the development of various urban functions, access to information and knowledge, mitigation of the negative environmental impacts, the protection of natural resources and cultural heritage, energy resource base development, sustainable tourism development incentives, and natural hazard impact limitation" [19] (pp. 244–311).

This research presents a case study, observing the phenomena within the context of the study area, which is limited to the Danube River in Serbia. In the literature review, the trend "of developing new journeys" was recorded, which gives context to the research presented here. Our research involves several activities, including reviewing existing data, such as statistical information from reports by river navigation authorities and development plans. It uses the results of previous analyses, such as those about the touristic profile, expansion of the river cruising industry and the spatial planning instruments, with the goal of finding a correlation between touristic product demands and the steps of the implementation of the strategy. In order to establish a trend and forecast future occurrences, the first step involves comparing data about the quantity and quality of cruising in the selected spatial segment. Using an integrated approach to planning sustainable development, particularly in relation to tourism, we include a balance between the growth and protection of natural resources and cultural values. For this reason, analyzing previous practice, and improving it in order to achieve a satisfactory level of alignment between the protection and development, required an explanation of the correlations of different levels of planning documents with respect to all restrictions and conditions. Considering the strategic goal, a scaling ratio that changes from the spatial plan level to the more detailed urban planning level can be used to analyze the most significant cultural products that are associated with cruising. The contributions here relate to overlapping tourist interests, sustainable development, protection conditions, technical requirements, resolving likely conflicts within the planning process, expert analyses, and public participation.

### 1.3. The Study Area

The Danube River is approximately 2850 km long and is the second longest river in Europe after the Volga River. The river originates in Germany, with inflows from 10 countries, and runs through four European capitals, namely Wien, Bratislava, Budapest, and Belgrade, before swerving through a delta in Romania and Ukraine and then into the Black Sea. Through Serbia, the river runs for 588 km

and the river's watershed covers 120 rivers, including some of the most important rivers in Serbia, including the Sava, Drina, Tisa, and Morava. It is a part of the extremely important Rhine–Danube European corridor and allows large cruises which bring many tourists to the Danube region, which is very profitable [20,21]. The European Union has recognized this corridor as one of the nine multimodal trans-European traffic corridors (TENT) and it is the only direct link between Central Europe and the Black Sea. The Danube is the only river in Serbia that is also a European corridor, thereby establishing the possibility of regional cooperation [22].

In the Podunavlje area in Serbia, significant attractions are located in vibrant urban centers such as Belgrade and Novi Sad, with seven fortresses from north to east downstream, including Bač, Petrovaradin, Beograd, Smederevo, Ram, Golubac, and Kladovo (Fetislam). There are also 21 archeological sites, the most significant of which are the Vinča and Lepenski Vir sites from prehistoric times, along with the city of Viminacium, Emperor Trajan's road, bridge and stone board with inscriptions from the Roman period [23]. Apart from these landmarks, two national parks (Fruška Gora and Đerdap), several other parks, special nature reserves, nature monuments, and areas with significant characteristics are situated in the surrounding areas and are protected areas [24]. The international EuroVelo 6 cycling route follows the river valley [25] and connects major cities with various attractions and events, along with historical places, nature reserves with rural settlements, wine and gastronomy routes, and other touristic options [26]. The Danube also features many river islands (53 in total), ten sand banks, unique locations at its widest point (about 6 km, between Moldova and Golubac) and at the narrowest sections of the river (Gvozdena Kapija/Iron Gates), as well as many suitable locations for marinas, piers, beaches, fishing, and bike paths. The lower Danube region is well known as the "historical zone of the Danube", featuring several attractive cultural monuments, including monuments from prehistoric times and ancient Rome. The region features a unique landscape formed by the penetration of the Danube between the Carpathian and Rhodope Mountains through Đerdap, creating Europe's largest and most attractive river gorge.

The details given above have been considered in the development of a strategic plan to develop and promote nautical tourism [27,28] along the Danube by providing new locations for passenger ports for cruises and marinas for individual boats. Nautical tourism has a high growth trend and the greatest increase is expected to occur in the Balkan Peninsula in Europe. Until 2020, forecasts have predicted that cruise businesses will be leaders in the global tourism industry. The large development potential of nautical tourism [29] in this region pertains to projects which involve increasing competitiveness for nautical products, marine tourist charter services, and river cruises [30]. The construction of nautical infrastructure along the Danube should contribute to a better position in Europe, increased tourism, and strengthened cross-border connections for Danube region tours and tourist offerings, thereby contributing to the economic development of the wider environment via nautical tourism [31–33]. The case studies and locations in Romania, as described by Gherasim and Gherasim, Irincu et al., and Matei et al., can be used for comparison with the present situation, as well as providing context for the strategic goals prescribed by the national spatial plan. International programs like "DIONYSUS—Integrating Danube Region into Smart & Sustainable Multi-modal & Intermodal Transport Chains" (http://www.interreg-danube.eu/approved-projects/dionysus), as a part of overall the Interreg Danube Transitional Program, have the task of assuring "a prospering waterborne transport sector, contributing to a sustainable transport system and regional growth. Besides better fairway conditions, a modern, energy-efficient fleet and better management of the transport system through comprehensive infrastructure planning and investment solutions are required. The investment needs refer to port infrastructure and superstructure, along with multimodal connections to remote ports". The priority is working towards better connections and energy use in the Danube region, and the specific objective is to support environmentally friendly and safe transport systems and balanced accessibility for urban and rural areas. Cooperation in such a project with partner countries is an opportunity to reconnect all key stakeholders and support cross-border linkages [34].

## 2. Persistence in the Creation of Tourist Products

In the past twenty years, Serbia has once again become an interesting location for travelers, not only as a city break location, but also because of the cultural and natural spots (mountains, wellness sites, and rural tourism) and numerous entertainment events (EXIT music festival in Novi Sad, Guča trumpet show, and Beer Fest in Belgrade, among others). The total number of foreign visitors in Serbia increased by 8% from 2018 to 2019; however, the unfavorable political and economic situation in years prior and the overall crisis in Serbia erased the country as a favorable destination and led to its negative perception. Research conducted in 2007 on the visitor experience when travelling to Serbia showed general satisfaction described, with positive comments related to the country's cultural heritage [35], hospitality, impressions about ways of life, and low cost [36]. The questionnaire was given to visitors at the EXIT music festival, visitors to Belgrade, and visitors who took international cruises on the Danube along pan-European transport corridor VII through Serbia. Event tourism is most popular among younger visitors, and the EXIT music festival, which is a well-established music festival with international character, takes place every summer in Novi Sad at the Petrovaradin Fortress (In 2019, EXIT attracted over 200,000 foreign visitors (Official EXIT Fest Site, https.www.exitfest.org) with around a 31% increase in comparison with 2018). Urban tourism is popular among young and middle-aged people and mainly takes place in Belgrade, which is the capital of Serbia. Cruise tourism is very attractive to visitors over 60 years old. For many years, only Novi Sad and Belgrade had proper docks for cruises, so the majority of visitors only had the opportunity to visit a small portion of the total cultural and historic heritage, where they only visited sites situated in those two cities [37,38]. On average, tourists spent four hours in Novi Sad and eight in Belgrade during their visit. Comparing the data from 2007, which is when the research about the impressions and satisfaction of foreign travelers to Serbia was conducted, and the data from 2019 (Table 1), this provides evidence of an increase in interest and consequently in profit.

**Table 1.** Number of foreign tourists in Serbia, statistical data from 2007 and 2019.

| Year | Total Number of Foreign Tourists in Serbia | Number of Cruise Passengers in Novi Sad and Belgrade | Number of Foreign Visitors in Belgrade | Number of Foreign Visitors at the EXIT Music Festival |
|---|---|---|---|---|
| 2007 | 696,045 | 445,183 | 296,461 | 50,000 |
| 2019 | 1,846,551 | 158,502 | 1,056,578 | 200,000 |

Sources: Tourist organization of Serbia [39], Port Governance Agency [40], EXIT festival official [41].

The "Strategy of Development for Water Transport in the Republic of Serbia 2010–2025" [42], in order to perceive opportunities for the future, compared Wien and Belgrade, both located along the Danube, which are similar in terms of their sizes and numbers of inhabitants (Table 2), where Wien shows seven times more cruise ship dockings and eight times more passengers. The conclusion of the report is that Belgrade should make better use of its unique position and tourist offerings since it is the most significant river port in Serbia. It is necessary to provide more opportunities with dispersed locations, cultural value, and products. Future port locations should be located near the main points of interest [43–46]. Profit regarding dockings and taxis is of important value, not only for services such as fuel supply or garbage disposal, but also in terms of marketing local goods and offerings such as excellent food or wine [47,48]. Moreover, this is significant as cruise tourists do not stay in hotels overnight and instead spend money at tourist sites and cruise stops.

**Table 2.** Comparative data about the ports in two capitals along the Danube: Wien and Belgrade.

| Achieved Results in 2014 | Wien | Belgrade |
|---|---|---|
| Number of citizens | 1,741,000 | 1,639,121 |
| Total length of the Danube in the country | 357.5 km | 588 km |
| Number of cruise ship dockings | 4036 | 510 |
| Number of cruise ship passengers | 460,265 | 68,000 |

Source: Danube Navigation in Austria 2014 and Strategy of Development of Water Transport in Republic of Serbia 2010–2025 [42].

## 3. Expansion of Danube River Cruises Until 2020

Although this paper was prepared during the COVID-19 pandemic, which has presented an enormous negative influence on the tourism sector, especially the cruising industry, the data from before 2020 demonstrate an increasing trend in travel and describe the potential for cruising along the Danube [49–51]. Based on this evidence, a strategy to develop more ports is clearly necessary.

Comparison of the annual numbers for dockings shows an increase of 65% in Belgrade's ports for a period of five years, along with 67% in Novi Sad, 45% in Gornji Milanovac, and even 95% in the new port of Golubac (Tables 3 and 4). In total, there were 1538 recorded dockings in 2019 with 208,644 passenger debarkations in Serbia recorded, with an increase of nearly 26% according to the 2018 data. In 2020, cruising stopped after the winter season in March due to the pandemic, where cruises and borders between countries were mainly closed. The Port Governance Agency recorded only five dockings by river cruisers in Serbia in 2020, with 408 passengers, presenting only 1% of those recorded in the previous season.

**Table 3.** Number of dockings in Serbian ports.

| Year | Belgrade | Novi Sad | Donji Milanovac | Golubac | Veliko Gradište |
|---|---|---|---|---|---|
| 2015 | 493 | 290 | 112 | 0 | 4 |
| 2016 | 531 | 306 | 112 | 0 | 3 |
| 2017 | 538 | 332 | 112 | 0 | 6 |
| 2018 | 587 | 347 | 143 | 44 | 21 |
| 2019 | 752 | 427 | 249 | 95 | 15 |

Source: Port Governance Agency, www.aul.gov.rs [40].

**Table 4.** Number of debarkation of passengers in Serbian ports.

| Year | Belgrade | Novi Sad | Donji Milanovac | Golubac | Veliko Gradište |
|---|---|---|---|---|---|
| 2015 | 65,494 | 35,314 | 13,673 | 0 | 203 |
| 2016 | 68,541 | 36,604 | 13,425 | 0 | 305 |
| 2017 | 72,279 | 42,622 | 15,562 | 137 | 625 |
| 2018 | 81,155 | 46,490 | 20,335 | 6225 | 2873 |
| 2019 | 103,523 | 54,979 | 34,260 | 14,607 | 1275 |

Source: Port Governance Agency, www.aul.gov.rs [40].

According to the data for the number of river cruises, the main season is during spring and summer, i.e., between April and October, and the peak occurs during May to September (Figure 1). Although the Danube is completely navigable in Serbia, the cold winter season periodically leads to ice forming on the river surface, making the river not suitable for river cruises. This corresponds with the seasons and arrangements for the international river cruising companies that operate in Serbia, with starting points in Germany, Austria, Slovakia, or Hungary. Interestingly, Serbia had an exclusive river speedboat "white fleet" until the late 1970s that has supported a tourist route between Belgrade and Donji Milanovac, which is a route that is approximately 220 km in length with a travel time of

approximately 3.5 hours. Memory of the fleet is still alive, but there has unfortunately been no success in trying to recreate it.

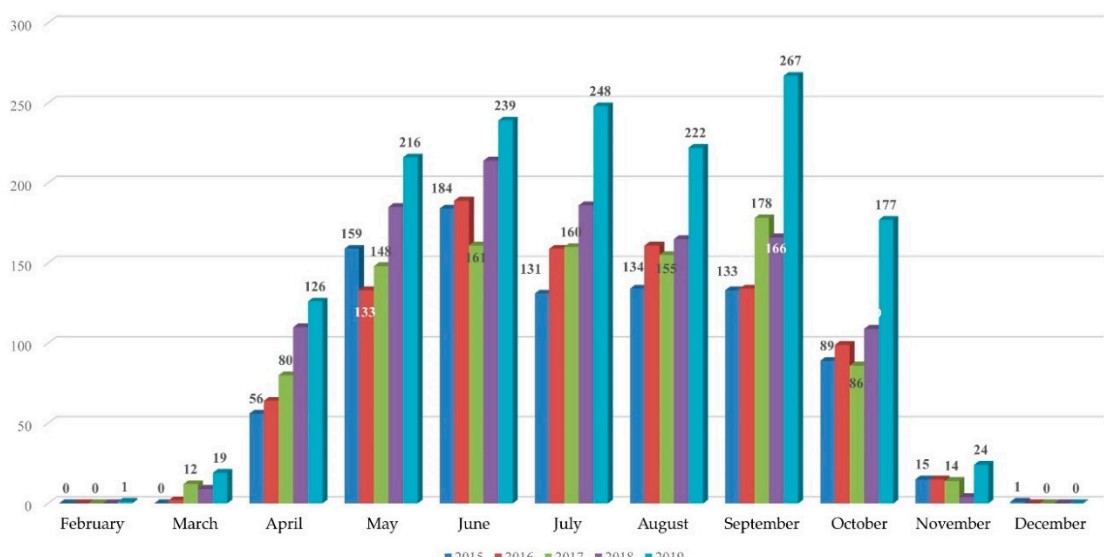

**Figure 1.** Comparative graph of dockings (*y*-axis: number of dockings), by months in the year, for the period of 2015–2019 (*x*-axis) (source: Port Governance Agency, www.aul.gov.rs) [40].

## 4. Implementation of Strategy through Spatial and Detailed Urban Plans

In order to perceive the needs and respond to the trend of river cruising, the "Strategy for the Development of Water Transport in the Republic of Serbia 2015–2025" [42] was developed in 2014, with the goal of establishing a basis for locating and planning new passenger ports not only along the Danube, but also the Sava and Tisa rivers. This strategy coincided with the Tourism Development Strategy of the Republic of Serbia 2016–2025 [52].

Apart from the existing docks in Belgrade, Novi Sad, Donji Milanovac, Golubac, and Veliko Gradište (Ram), several more are planned along the Danube in Apatin, Bačka Palanka/Banoštor, Sremski Karlovci [53], Zemun, Smederevo, Kostolac, Lepenski Vir, Kladovo, Negotin. More docks are planned along the Sava River in Sremska Mitrovica, and Šabac. The Sava River runs from the west, starting in Slovenia, then flowing through Croatia and Belgrade, then flowing into the Danube. Additionally, more docks are also planned along the Tisa River in Kanjiža, Senta, Bečej, and Titel. The Tisa River flows from the north, starting in Ukraine, then flowing though Romania, Slovakia, and Hungary. In an altitudinal sense, Belgrade's passenger port is located below Belgrade's fortress and old city center and is actually located on the shoreline of the Sava River. A new port in Zemun will be close to it, with the purpose of relieving traffic pressures in the existing port. The new port is expected to provide a similar touring experience for touring around the Belgrade metropolitan area. The other location is Vinča, which is a village located approximately 12 km downstream from the city center, where the archeological site "Belo Brdo" (also known as "White Hill") is situated and features Neolithic remains.

Details regarding the planning documents in Serbia are given in Table 5, detailing the basic characteristics, content, and determinations of the chosen plans for the case study area. The Spatial Plan for the Republic of Serbia from 2010 to 2020 [54], as the new version of the plan for 2021–2035, which is now in the finalization phase before official adoption, recognizes the potential tourism in Podunavlje and suggests that an integrated approach for planning, gathering touristic products, and maintaining cultural heritage, including river cruising, continues. The plan remarks that "waterways are only partially used in relation to their potential capacities. Domestic traffic is not connected to the navigation roads in the Rhine and Danube systems, so domestic ports and docks have not been rebuilt, nor has

the fleet. The main problem is the negative effects of the privatization of ports, which can abolish port activities and cause the Republic of Serbia to lose its strategic position on the Danube".

**Table 5.** Details of the planning documents.

| Planning Document | Relevance to the Case Study | Characteristics, Content, and Determinants |
|---|---|---|
| Spatial Plan of the Republic of Serbia | The overall document | The economy, transport, tourism, cultural co-operation, and other forms of connectivity along this development axis will support plans and projects related to corridor VII, i.e., a joint development strategy based on interstate co-operation between Danube states and the region. |
| Spatial Plan for the Special Purpose Area (SPASP) | SPASP of the International Waterway E 80 (i.e., the Danube or pan-European corridor VII) | According to this plan, existing tourist product categories are manifestation, hunting, fishing, city and health tourism (spas and wellness locations) and tourist products with exceptional potential. Nautical and ship cruising, special interests, rural tourism, events, and other categories are underregulated and affirmed. For tourism development, the international waterway has significance for international cruise ships and also for boats of all categories. |
| Detailed Urban Plan (DUP) | DUP for "Dragulj" in Kostolac | The plan includes land use for cruise ship ports with customs offices, tourist information spots, souvenir shops, cafés, car rental, parking for individual cars and tourist buses, marinas with category 3 anchors, small airports for sport airplanes, parks and recreational areas, sport polygons, camping sites with facilities, bicycle paths, and hotels and apartments. |
| | DUP for "Marine Smederevo" in Smederevo | The plan considers several aspects, including marina management offices, technical maintenance and overhauling vessels (with a crane for extraction), ports/docks for large cruise ships on the Danube, commercial uses (restaurants, shopping, and services), sailing schools, indoor pools, hotels, and outdoor sports and recreation complexes. |

The Spatial Plan for the Special Purpose Area of the International Waterway E 80 (i.e., the Danube or pan-European corridor VII) is a key planning document and was adopted in 2015 [55]. The implementation of its goals is possible by developing more detailed urban plans for specific locations (Figure 2). All the aforementioned documents are coherent in their recommendations for passenger port locations. These recommendations are based on investigations of the most attractive cultural offers and natural goods and values. The assets of the Danube, from the point of view of tourist valorization, are important in terms of the rich tourism motives, ecological preservation aspects [56–58], and the natural and cultural heritage offers. These assets are complemented by coastal cities and partly by general travel accessibility.

Prior to 2019, several detailed urban plans were adopted, including plans for passenger ports in Kostolac [59] and Smederevo [60]. The principles of the plans have been different and some locations are new, such as Kostolac; however, the Smederevo plan still intends to reuse reconstructed industrial docks. All locations are developed according to departmental law [61] and other acts regarding the given technical conditions and requirements.

In Smederevo, the area of the plan directly borders with the Smederevo Fortress, which was declared as a cultural site in 1975. In 2013, with the decision to determine immovable cultural goods with exceptional importance ("The Official Gazette of the SRS", no. 14/79), the Smederevo Fortress was categorized as a cultural good with exceptional importance to the Republic of Serbia.

The plan for Kostolac near the thermoelectric power station occupies an area of nearly 53 ha, directly on the Danube shore, with two industrial canals on both sides. This location is located about 5 km from Viminacium, which is a significant Roman archeological site that was previously a settlement

and military camp. Additionally, the city of Požarevac is nearby, which features museums and art galleries, and the stud and horse racetracks in Ljubičevo are also noteworthy.

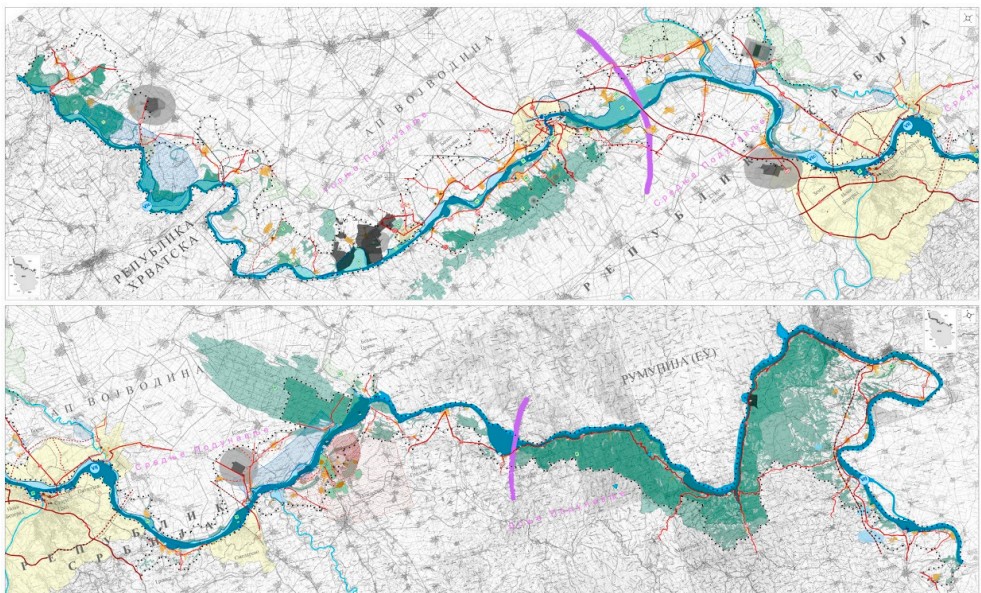

**Figure 2.** The Spatial Plan for the Special Purpose Area of the International Waterway E 80 (i.e., the Danube or pan-European corridor VII). Natural goods are indicated by a green color, with waters in blue, tourist region borders in violet, and settlements in yellow. Data sourced from the Spatial Plan for Special Purpose Areas [55].

The two examples given in Figure 3 highlight the idea in the spatial planning to investigate and find the most suitable locations on river banks along the Danube to organize and build passenger ports. These ports would be built next to or near to cultural heritage complexes and provide facilities to support pleasant visits.

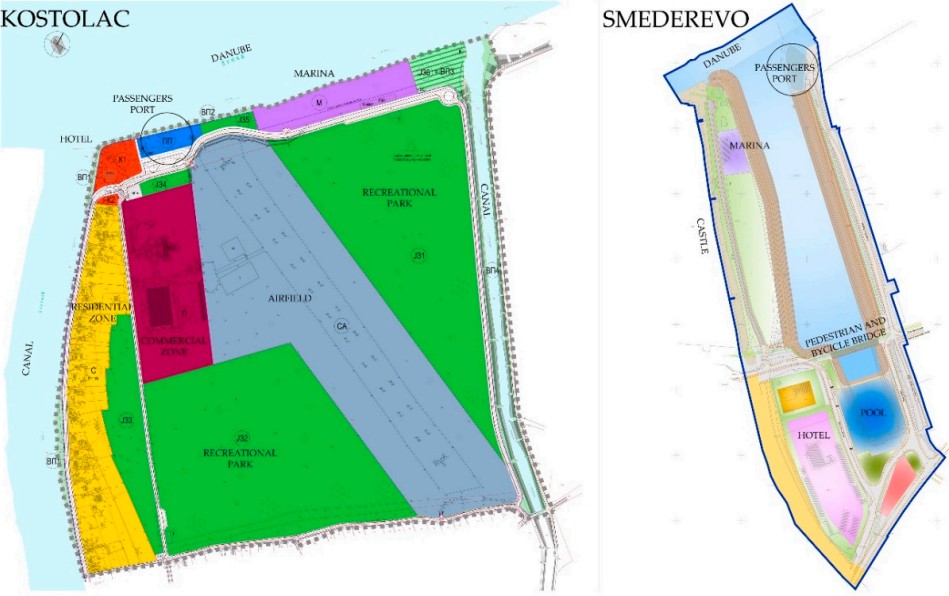

**Figure 3.** Detailed urban plans for passenger ports in Kostolac (left, data from the Detailed Urban Plan (DUP)/public document, Institute of Architecture and Urban & Spatial Planning of Serbia) [59]) and Smederevo (right, data from the DUP/public document, Municipality of Smederevo) [60]). Land use maps.

## 5. Heritage Locations as a Factor for Locating New Port Locations along the Danube

In order to present as much as possible during short visits and save time for the organization of field trips to remote areas, possible port locations have been chosen close to cities [62,63] and areas with cultural or natural protection characteristics and significance (Table 6). The historical and geographical relevance of the Danube water corridor presents excellent circumstances for tourism, with many cities, fortresses, remains of prehistoric and Roman settlements, churches, and other notable locations close by or directly on the river's banks (Figures 4 and 5). It is possible to enjoy natural beauty from the deck of a cruise ship, and cruise ship tourists are not interested in spending time in nature (i.e., camping, hiking, or skiing), but instead prefer comfort with short and well-organized breaks with opportunities to see, learn about, and experience new locations and goods. Leisure time is commonly spent gaining knowledge, and visiting and learning about cultural heritage sites to make great memories and keep or share attractive pictures have become essential activities for modern travelers. Decisions regarding potential port and cruise ship stops are made on this basis, but it is also necessary to consider the required technical conditions regarding the characteristics of the shore, land use on the banks, and connections with the road network, and most of all the safety of docking and navigation. If the surrounding area of a stop is not attractive, only has a local identity, is overly complicated to visit, or has some problems with fulfilling technical conditions, then the location is not suitable for tourism. Such locations would require further analysis and should either be moved to a more suitable location, made more convenient, or omitted entirely.

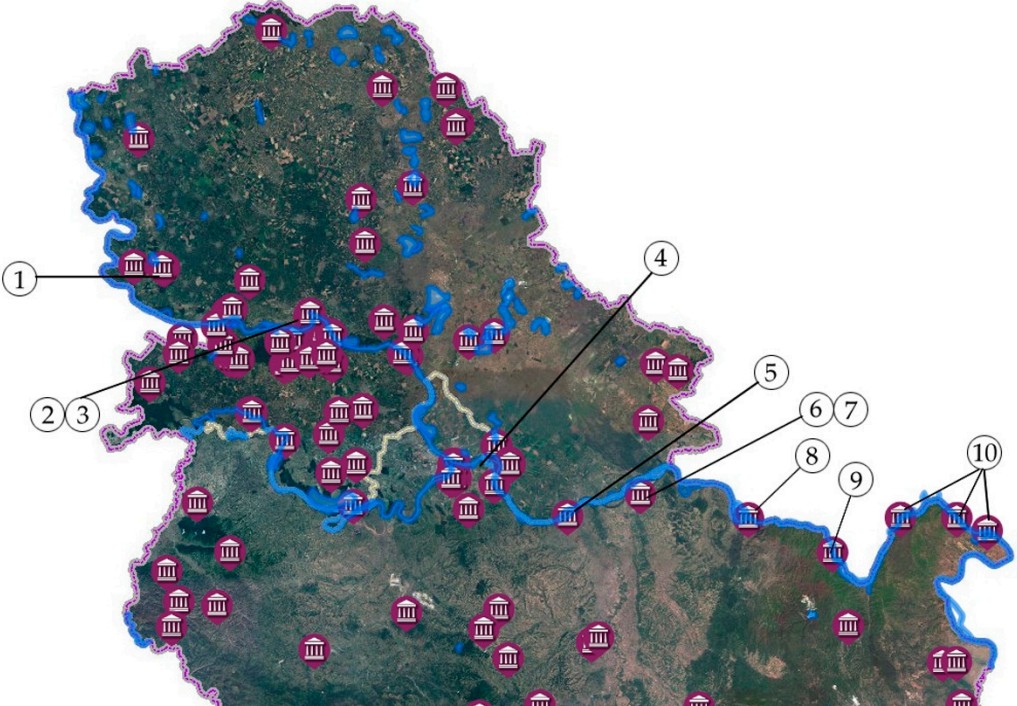

**Figure 4.** Danube corridor with locations with protected cultural heritage and nearby planned positions for ports (1—Bač; 2—Novi Sad, Petrovaradin, and Fruška Gora; 3—Sremski Karlovci, 4—Belgrade, Zemun and Vinča, 5—Smederevo, 6—Kostolac, Viminacijum, 7—Ram, 8—Golubac, 9—Lepenski Vir, 10 —Kladovo, source: https://geosrbija.rs [64]).

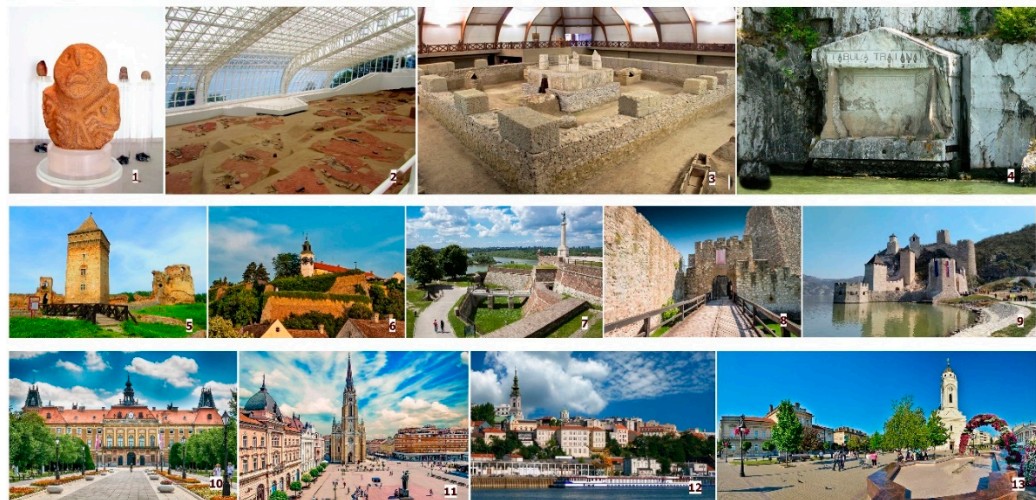

**Figure 5.** Various cultural heritage sites in Podunavlje, including archeological sites (top row: 1. Danubius sculpture, 2. Lepenski Vir, 3. Viminacijum, 4. Tabula Traiana), fortresses along the Danube (middle row: 5. Bač, 6. Petrovaradin, 7. Belgrade, 8. Smederevo, and 9. Golubac), and urban centers (bottom row: 10. Sombor, 11. Novi Sad, 12. Belgrade, and 13. Smederevo). Source: Touristic organization of Serbia and public Internet presentations [39].

Interestingly, there are mutual gains because of the increased interest in cultural heritage at some sites (e.g., Golubac and Lepenski Vir). Some sites have recently been reconstructed and their presentation has been improved to contemporary standards and needs by adding visitor centers (Viminacium) or additional events and performances (Viminacium and Gamzigrad-Romuliana). It is now possible to not just to visit and see heritage sites, but also to understand what they were like in the past and to experience simulations of life in the Roman Empire (e.g., to see soldiers dressed in uniforms with traditional arms, travel in two-wheeled vehicles, taste food and wine prepared by following original recipes, listen to music from the period, etc.). It is also feasible to create prehistoric or medieval simulations.

Of the 13 planned ports, six of them currently exist. In terms of the planned additions, three ports are in the process of realization and four more are planned (Table 6). Most distances between ports are approximately less than 20 km, which allows them to be used as alternative ports for several attractions and to reduce crowding. Ports located between 25 and 55 km apart make it possible to combine two or three stops and visits per day or to choose one of them. Only four ports have distances of 75–80 km between stops, which is ideal for cruising and enjoying time on board the cruise ship. In terms of the distances between planned ports and accessibility to cultural locations, four port locations require bus transport since they are between 5 and 30 km away from attractions, and some are around 65 km away from heritage sites and cultural products. Six ports in urban centers of different sizes have the advantage of being close to points of interest and make it possible to combine sightseeing on foot with bus tours. Three ports are in ideal locations next to attractions and are accessible by foot. This is essential for planning visits and excursions when travelling via a boat, where the accessibility allows sufficient time to fully enjoy the location and see everything on offer. Six existing ports and three in the process of preparing documentation or buildings are located in areas that provide fast and easy access to nearby attractions. Only one of the planned locations in Lepenski Vir is next to an archeological site, and the rest of the planned ports are approximately 30 km away from sites, making it a necessity to organize transport for passengers.

**Table 6.** Passenger port locations and nearby cultural products.

| Port Location and Status (Existing/Suggested or Planned/In Realization) | Nearby Heritage Sites and Cultural Products | Location Characteristics | Distance from the Port and Type of Required Transport to the Spot |
|---|---|---|---|
| Apatin (planned), with about 75 km to the next stop | City of Sombor | Ambiental value and an immovable cultural good with exceptional importance. Supports walking and visiting. | 20 km by bus |
| Bačka Palanka and Banoštor (on opposite sides, planned), with about 40 km to the next stop | Bač Fortress, monasteries in Fruška Gora National Park, and a wine route | Ambiental value and immovable cultural goods with exceptional importance. Supports walking, visiting, and tasting. | 20–30 km by bus |
| Novi Sad (existing), with about 15 km to the next stop | City of Novi Sad, Petrovaradin Fortress, monasteries in Fruška Gora National Park, and a wine route | Ambiental value and immovable cultural goods with exceptional importance. Supports sightseeing and tasting. | 0–25 km on foot or by bus |
| Sremski Karlovci (in realization), with about 80 km to the next stop | Old city, nearby to Petrovaradin and Novi Sad, monasteries in Fruška Gora National Park, and a wine route | Ambiental value and immovable cultural goods with exceptional importance. Supports sightseeing and tasting. | 0–25 km on foot or by bus |
| Belgrade, Zemun, and Vinča (existing/planned), with about 65 km to the next stop | City of Belgrade, Klamegdan Fortress, old city core (pedestrian zone, bohemian quarter, Skadarlija museums (including Museum of Nikola Tesla), and a Neolithic archeological site in Vinča | Ambiental value and immovable cultural goods with exceptional importance. Supports sightseeing, walking, gaining knowledge, and exploring. | 0–15 km on foot or by bus |
| Smederevo (in realization), with about 25 km to the next stop | City of Smederevo, a medieval fortress, and a wine route | Ambiental value and immovable cultural goods with exceptional importance. Supports sightseeing, gaining knowledge, and exploring. | 0–15 km on foot or by bus |
| Kostolac (in realization), with about 40 km to the next stop | Cities of Požarevac and Ljubičevo, archeological sites, and Viminacium | Ambiental value, immovable cultural goods with exceptional importance, and a special purpose area. Supports visiting, gaining knowledge, and exploring. | 5–15 km by bus |
| Veliko Gradište (existing), with about 20 km to the next stop | Ram Fortress | Ambiental value and an immovable cultural good with exceptional importance. | 0 km on foot |
| Golubac (existing), with about 55 km to the next stop | Medieval fortress and Đerdap National Park | Immovable cultural goods with exceptional importance. Supports gaining knowledge and exploring. | 0 km on foot |
| Donji Milanovac (existing), with about 15 km to the next stop | Emperor Trajan's road and board from the Roman period, archeological sites, Lepenski Vir, and Đerdap National Park | Immovable cultural goods with exceptional importance. Supports walking, gaining knowledge, and exploring. | 0–15 km on foot or by bus |
| Lepenski Vir (planned), with about 80 km to the next stop | Prehistoric archeological sites | Immovable cultural goods with exceptional importance. Supports gaining knowledge and exploring. | 0 km on foot |
| Kladovo (existing), with about 55 km to the next stop | Fetislam Fortress, Trajan's bridge, Karataš, and Diana Fortress | Sites from the Roman and Byzantine periods and immovable cultural goods with exceptional importance. | 0–15 km on foot or by bus |
| Negotin (planned) | Archeological sites, Gamzigrad-Romuliana (a UNESCO site), and the Rajačke Pimnice | Immovable cultural goods with exceptional importance. Supports gaining knowledge, exploring, and tasting. | 30–65 km by bus |

## 6. Discussions

As Tomej and Lund-Durlacher concluded with their parallel comparison of the characteristics of ocean (sea) and river cruises, and given the research conducted by numerous authors, there are many differences between the two modes, especially in terms of the volumes and corresponding entities. It is important to identify the differences between mass tourism (associated with charter packages, hotel resorts, or sea cruises) and the river cruise industry (Figure 6) and to establish an appropriate balance. This is necessary to assure more ports in proximity to different cultural places, equalize schedules, define maximum numbers for bookings to present heritage sites appropriately, protect areas, and also to satisfy the need for revenue, as economic gain is an important input for future maintenance and development. River cruises should be orientated toward enjoyment, expanding knowledge, and experience, tasting, and following the slow flow of the river. Historically, this way of traveling started as an exclusive method of travel and later became affordable for the wider population. The strategy to increase the number of ports and construct them close to points of interest has the obvious goal to take advantage of this relatively new and still unsaturated market in order to increase income; however, on the other hand, the strategy has a more important target—to promote culture heritage in order to build deeper connections within the region and Europe overall.

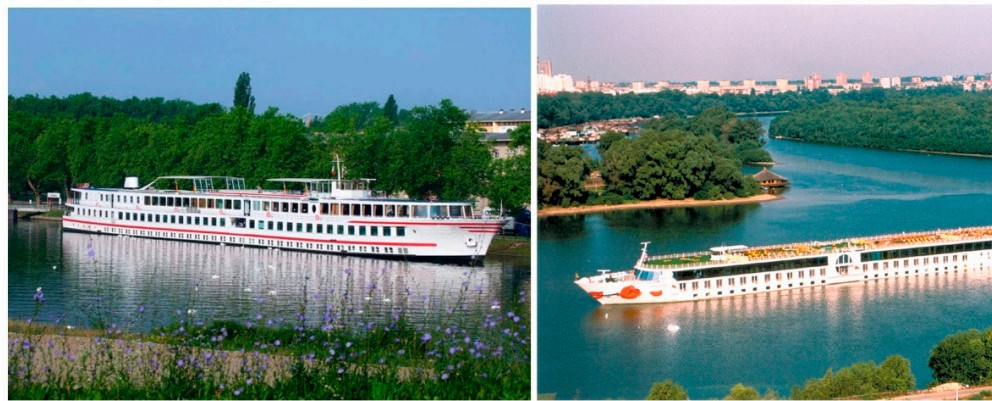

**Figure 6.** River cruises along the Danube River in Serbia. Source: http://www.podunavlje.info [65].

Two aspects were crucial in terms of choosing potential locations and criteria for strategic planning, namely the distances from cultural products and basic technical conditions. On a more detailed level, suitability was elaborated on in the form of the assessment of impacts and by obtaining spatial conditions from relevant institutions responsible for environmental and heritage protection. With the combination of overlapping layers of protection zones and defined restrictions, along with all other technical details, it is possible to locate ideal positions for ports. The process of spatial and urban planning has an important component for participation in the form of public insight and discussion. Citizens are usually interested in plans that concern their private properties; however, in this case, when the topic is the public good of river banks, opinions about projects may also be very important. In local cases, there were no opponents to the proposals, because the development of tourism is still in the sphere of positive and progressive measures. As McCarthy stressed in the case of the Valletta Waterfront Project, besides "clear economic benefits, there are associated problems that may become evident in the medium- or long-term. The resulting issues and tensions indicate the need to more sensitively evaluate cruise tourism-related development proposals" (pp. 341–350). For that reason, it will be necessary to monitor not only the implementation and impacts on the environment, but also the local community in order to compare the aspects of improvements that locals can also enjoy. Both detailed plans used in the case study consider not only project proposals for establishing conditions for tourist port locations, but also combined land uses and facilities with public interest, and this should be taken as a positive example.

The strategy highlighted here is a positive in idea in terms of developing and promoting tourism offers using the Danube as a main axis, although the remarks here regarding the results before finalization are tentative, with the belief that the number of suggested locations is sufficient and should not be expanded, except in cases where potential locations are evaluated as impossible after more detailed consideration. The obligation to involve all important stakeholders and collect, compare, and harmonize their interests and requests is a regular part of planning. Until now, this has been the leading measure to ensure the realization of tourist and water transport strategies in Serbia. The authors believe that established plan hierarchies and institutionally led processes are guarantors for realization without obstructions and negative consequences for the environment, local society, or protected areas.

## 7. Conclusions

In this study, the importance of developing tourist offerings in the Podunavlje region in Serbia and investing in locations for passenger ports has been highlighted via a review of scientific articles and several strategic documents and plans. Although being negatively influenced by the COVID-19 pandemic, with river fleets presently remaining anchored in European ports with no prediction of when it will be possible to travel again, it is expected that, in the future, the conditions will normalize and the increasing trend in the number of cruise ships will continue. Knowing whether international cruise companies and fleets will survive this difficult period is impossible to predict. This may be a chance to develop domestic lines that will operate along the Danube and fulfill local tourism demands instead. Short trips and excursions in the Belgrade city area or to Vinča, Golubac, or Donji Milanovac could be expanded and extended with a focus on cultural points of interest. On the other hand, this is an opportunity for new research after the pandemic to perceive and calculate the effects of COVID-19 and keep track of the recovery and progress. As a result of the pandemic, there may be some changes in terms of the organization and abundance of tourism.

While expecting normalization and a return to previous trends in this uncertain period, it is reasonable to continue with planning and realization in terms of preparing for future tourism seasons. Connections within the region and cultivating cross-border collaboration can even make tourism offerings richer, thereby creating a better distribution of tourists in destinations. It is very important to keep decision-making about new ports on the governmental or strategic level with the certain perception of overall need and possibilities, considering the region not only as a part of Serbia, but also viewing it from the perspective of cross-border cooperation. On this level, it is necessary to determine all different and overlapping sectoral views and policies, starting with tourism, ecology and sustainability, protection zone restrictions, resolving potential conflicts and defining positive aspects. On the other hand, based on strategical conclusions, it is possible to delegate the implementation to the local authorities.

With several suitable ports, cruise and excursion packages could be increasingly diversified to focus on diverse locations and themes. This could make it possible to attract not only elderly passengers, but also young people, professionals, and families with children, or even to include the same visitors in several cruise trips so that they can experience everything the Danube has to offer.

The limitation of this research is the lack of similar case studies for comparison, since the available research commonly pertains to seaports or highly developed Western and Central European river ports. Few academic papers have elaborated on the Danube. A paper considering Romania came to similar conclusions as the present study, where the development of passenger ports should be close to points of interest, that regional cooperation and connectivity is important and that there is a need to establish local cruising lines as an alternative version of cruising. The current challenge is the fact that developing and opening new ports in Serbia is still in progress, so there is still no clear evidence in terms of the finalization status when considering the project goal. This is why we must follow and monitor the progress and impacts in the future. It is feasible to seek increased interest in the region considered here; however, because of its peripheral location, it is difficult to expect that tourism traffic will extend to the upper Danube.

Based on our analysis, comparison of tourist needs, and protection requirements from the point of view of sustainability, this paper can serve as a basis for further research. This could include studies on economic impacts, resolving the spatial overlapping of protection zones and tourist areas of interest, tourist behaviors and satisfaction, creating products and adjustments for different passengers, and, potentially as the most important direction, considering sustainability and critical levels of exploitation. A necessary precaution for this is to collect as many different quantitative data about the topic as possible in order to observe locations and processes from many perspectives and to involve all stakeholders (i.e., tour operators, port authorities, local municipalities, tourist organizations, tourist passengers, experts in heritage and environmental protection, and citizens). Finally, it is important to analyze the preservation of economic assets and the protection of heritage sites and offers, and additionally to measure the promotional influence of targeted cultural and tourist products.

**Author Contributions:** Conceptualization: N.D.H.; methodology: N.D.H. and N.S.; formal analysis: N.S.; investigation and resources: N.D.H.; data curation: N.D.H.; writing—original draft preparation: N.D.H. and N. S.; writing—review and editing: S.M.; visualization: N.D.H.; supervision: S.M. All authors have read and agreed to the published version of the manuscript.

**Funding:** This research received no external funding.

**Conflicts of Interest:** The authors declare no conflict of interest.

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
