# Peer review of "Danube River Cruises as a Strategy for Representing Historical Heritage and Developing Cultural Tourism in Serbia"

_sustainability, doi:10.3390/su122410297_

Round 1

Reviewer 1 Report

The subject is very interesting. River cruises are an under-researched topic in tourism literature. The present paper isn’t methodologically innovative, however, it’s interesting for the scientific community as a case study related to Serbia and one river Danube.

The paper should be reviewed and restructured. The subtitles and the contents should be configured as usual in an article, paying special attention to the methodology and research technique used, and to present the discussion section.

Review the English very carefully.

ABSTRACT

The abstract should be an objective presentation of the article. The abstract should be reviewed according to these guidelines: (i) highlight the purpose of the study, isn’t clearly defined – what is the main goal?; (ii) describe the main methods or research technique, the reviewer suggests that the authors choose the case study, it must be an option to restructure the paper; (iii) summarize the article's main findings, indicate the main conclusion. One statement like this, line 27: “The result of the research is defining the policy and the concept of sustainable tourism development along the Danube river in the wider regional” is very broad and ambitious for an article. The authors don't actually do this. They don’t define the policy and they don’t explore the concept of sustainable cruise tourism development on this destination. 

INTRODUCTION

The lack of theoretical part is evident. The paper starts with the introduction of the case study. The key to this section is to provide a theoretically reasonable conceptual framework about cruise tourism. The reviewer suggests including a few academic bibliographic references related to the background of scientific discussions in the international literature about cruise’s tourism, namely river cruises. The systematization of the literature review must be in line with the main theoretical topic of the paper, objectives of the study, with orderly writing from the general to the particular. The paper doesn’t provide sufficient background information and literature review regarding these topics: river tourism development, the increment of the river cruise industry in general, before the case study. The reviewer suggests that the state of literature about fluvial tourism, cruise tourism, fluvial planning, should be done before the case study. The current state of the research field should be reviewed carefully, and key publications cited. 

See, amongst many other references related to river tourism and/or river cruises:

Dwyer, L., & Forsyth, P. (1996). Economic impacts of cruise tourism in Australia. Journal of Tourism Studies,7(2), 36–45.

Dwyer, L., & Forsyth, P. (1998). Economic significance of cruise tourism. Annals of Tourism Research25(2), 393–415.

Hilma Tamiami Fachrudin, H. T., & Dolok Lubis, M. D. (2016). Planning for riverside area as water tourism destination to improve quality of life local residents, case study: Batuan – Sikambing River, Medan, Indonesia. Procedia - Social and Behavioral Sciences234, 434 – 441. Procedia - Social and Behavioral Sciences   234  ( 2016 )  434 – 441.doi: 10.1016/j.sbspro.2016.10.261 

Lund-Durlacher, D., & Kristof Tomej, K. (2020). Research note: River cruise characteristics from a destination management perspective. Journal of Outdoor Recreation and Tourism, 30, https://doi.org/10.1016/j.jort.2020.100301

McCarthy, J. (2003). The cruise industry and port city regeneration: The case of Valletta. European Planning Studies11(3), 341–350.

Prideaux, B., & Cooper, M. (2009). River tourism. Oxfordshire: CABI Publishing.

Van Balena, M., Michael Doomsa, M., & Elvira Haezendoncka, E. (2014). River tourism development: The case of the port of Brussels. Research in Transportation Business & Management, 13, 71-79. 

METHODOLOGY

The methodological frame presented is very ambiguous. On line 72: “the methodological frame used in the research regards to multidisciplinary approach combining knowledge form fields of tourism, protection of public goods, preservation and presentation of the cultural heritage and spatial and urban planning. This integrated method leans to the context of sustainability, with assessment of benefits and disadvantages, observe and harmonize spatial demands”, the reviewer suggests that the authors should set a specific methodology and research technique. As was referred to in the abstract, the case study could be an option. In the opinion of the reviewer the case study is the scientific research technique used but not explicitly assumed.

As a suggestion, what the authors think about creating a subsection titled “study area”?.

MAIN TEXT

The text is sometimes overly descriptive. Some information like plans could be systematized in one table. Interpretation and explanatory approaches are more interesting.

The section “4. Implementation of strategy through the spatial and detailed urban plans” is sometimes very descriptive; changes must be made. 

One question, do the authors really agree with this statement? Line 35: “In addition, water traffic is environmentally cleanest and safest with numerous development opportunities.” When the authors look to the photos presented in line 296 they really think as they write about “environmentally cleanest”?

Rethink subtitle 2., line 82: ‘’It doesn’t matter where I’m from, as long as you know where I’m traveling’’ isn’t useful in the paper. This subtitle says nothing about the content. Changes must be made.

Pay special attention to situations like that, line 166: “The Belgrade’s passengers port is located beneath”. The authors must be using a geographical reference (with cardinal points or upstream or downstream). Take this aspect into account throughout the article.

CONCLUSIONS

It is necessary to contrast the case study with the state of the literature. 

Line 281 “It is important to make difference between mass tourism and river cruise industry”, the reviewer asks: are river cruises not getting massified? This statement must be questioned, discussed, and not made an uncritical statement. What is happening in the Danube? How to make this activity sustainable at this destination, considering the multidimensional nature of sustainability? What kind of measures has been taken at this destination? Considering the title “strategy for representing the historical heritage and developing of cultural tourism in Serbia”, the reviewer would like to ask what are the real strategy? What is the critical position of the authors in relation to this strategy? Who are the most important stakeholders? On line 285, the authors refer to the importance of “resolving conflicts”. What kind of conflicts exist considering cruises tourism at this destination? This could be the core of the discussion section.

The discussion section can be separated from the conclusion section. The authors should consider presenting separately, these two sections: “Discussion” and “Conclusion”.

Although the word discussion is used “discussion and conclusion”, the authors do not actually discuss.

Conclusion: concentrate a bit more on current challenges and future perspectives.

The limitations of the study and possible future research due to this theme are missing. Something should be written about this.

FIGURES

Some Figures don’t have quality, for example, Figure 1, Figure 2, Figure 3, and Figure 4, are not readable, some of them hold very low quality, low dimension, and low resolution. The maps should have more legibility. 

Figure 1 must be presented in English. 

Figure 2 Caption has no reading.

Figure 3 Very small without reading.

Figure 4 Very small without reading.

Figure 5 Identify all cultural heritage in Podunavlje, presented in the figure. 

The sources and credits of the images must be mentioned.

Line 296 two figures without title and source. These two figures aren’t mentioned in the text. The opinion of the reviewer is that these figures should appear in the discussion, not in the conclusion.

TABLES

The contents of table 5 aren’t interpreted in the text. The main conclusion should be present, namely refers to the distance mean considered, the balance between existent and planed, amongst others.

Author Response

Thank you for review, responses are in the attachment.

Authors

Reviewer 2 Report

Title:Danube’s river cruises as a strategy for representing the historical heritage and developing of cultural tourism in Serbia

#1 The article is entitled “Danube’s river cruises as a strategy for representing the historical heritage and developing of cultural tourism in Serbia”. It is focused on reviewing the impacts of strategic spatial planning on the location of river ports and marinas along the Serbian Danube in view of the promotion of cultural tourism in Serbia. The article argues that the location of heritage sites has been a crucial factor on the decision of the location of the new Danube’s ports. The article aims to provide useful contributions to the definition of a “policy and concept of sustainable tourism development along the Danube river in the wider regional context of Southeast Europe”.

#2 While the paper offers a potentially interesting case study, I feel that it lacks several steps to make it a sound and consistent scientific study and research.

# 3 First, a clear argument and conceptual contribution are missing. The majority of the text is largely descriptive about the Serbian Danube river and valley, providing for quantitative data and a description of the main strategies and plans for the area. However, the current draft does not make the research question and argument clear. On the contrary, it is quite difficult to identify what exactly are the objectives and goals of the research and what is the main theoretical discussion behind the presentation of the case study.

Is it the discussion on trans-national planning regions and the need for a joint-up thinking and cross-border coordination? I think this might be a very interesting discussion to which the case of Danube perfectly fits. However, this has not been done.

Is it, on the contrary, the relationship between spatial planning and sustainable tourism development policies settled over the discussion on the location of river ports and the cruising industry? If so, this has not been done either.

In short, a clear argument and theoretical discussion are missing. Besides, in my opinion, the simple reviewing of passengers’ ports location on Serbian Danube in view of nearby heritage  is too short for a scientific paper.

#4 Second, the methodology used is also not clearly explained. In the introduction, there is a brief note to the adopted methodology. It is mentioned that the methodological framework makes use of a multidisciplinary approach that combines different knowledge fields, ranging from tourism and cultural heritage to spatial and urban planning. However, aside from this brief note in the introduction, no other explanation exist on the methodology, how is it related with the conceptual framework, which steps does it comply with, how are these steps articulated, etc. Moreover, there is no evidence on how is this methodology leading to the final goal of the research: contribute to the definition of a “policy and concept of sustainable tourism development along the Danube river in the wider regional context of Southeast Europe”. Clarification on the methodology is needed to better understand the given data and the analysis done of each planning instrument.

# 5 Apart from the strategies and plans themselves, literature on the issues of spatial and urban planning is almost absent, when this is an important part of the research that relies on the planning strategies and instruments at regional and local level.

# 6 The overall organization of the paper is not presented in the introduction and it becomes too difficult to understand how the several parts are related and how they are contributing to the global rationale.

# 7 Discussion and conclusions are too short for a sound scientific paper. They came to introduce the pandemic issue while none of this has been addressed in the article and no data regarding the impacts of COVID-19 have been collected. New concepts and discussion (e.g mass tourism vs river cruise industry) are introduced that have not been pinpointed in the literature review.

# 8 The title gives little insight on the background discussion. It is mainly focused on the case study itself and does not reflect any theoretical insight.

# 9 The article needs an extensive English proof-reading.

# 10 To conclude, I think major structural revisions are needed to make this article acceptable for publication.

Author Response

Thank you for the review, responses are in the attachment.

Authors

Reviewer 3 Report

I think that river cruises is neglected for cultural tourism. In this case conected river cruises and cultural tourism.

Author Response

Thank you for the review, the responses are in the attachment.

Authors

Reviewer 4 Report

The abstract should be rewritten. 

The research objectives and main results aren't clear and most researchers start their readings by the abstract, introduction, and conclusion sections.

Authors shouldn't use the word "etc"  line 87. Instead should use "among others".

Conclusions should be improved. Figures shouldn't be used in the conclusions section.

The references section is very extensive but is not reflected in the literature review. The literature review should also be improved, namely regarding cultural tourism and experience co-creation.

Author Response

(The authors gave the same response as above.)

Round 2

Reviewer 1 Report

The abstract should include a reference to the methodology: case study.   Review the expression et al.   Line 235: removes the point (Table 1.).   Figure 5 Identify all cultural heritage in Podunavlje, presented in the figure, using letters or numbers for each figure.   Figure 6, insert the source.  

Line 355  (right, , resourse): removes the comma; resource.

Pay attention: "Resourse", "resource".

Reference should be made to the source, in all figures and tables use the word "source" not "resource".   Figure 1 - Identify the units in the x-axis.   Tables insert the source at the bottom.

Reviewer 2 Report

#1 The paper has been greatly improved in regard to the first version. A new theoretical framework on river cruise tourism has been added, introducing an important discussion on the differences between ocean and river cruises and raising attention on the opportunity of such a study and research. Additional elements also include a section dedicated to the methodology and a table establishing the relationship between the several spatial planning instruments. Significant improvements were also made in the discussion. However, I feel that there are still some aspects that haven’t been fully explored and that might deserve an extra attention.

#2 Methodology: Although the paper is now providing for a methodological section – “Methodological Framework and research Technics”, I think this is still not clear. In this section, authors mention the “multidisciplinary approach” and the “integrated method”. However, these correspond to some tips about the methodology spatial planning instruments themselves have followed, instead of presenting the research methodology. As such, the research evidence that a number of analysis have been made (e.g. analysis of the touristic profile; analysis of the spatial planning instruments; etc.) but no previous explanation is given the several methodological steps and their correlation.

#3 An important part of the research is focused on the way spatial planning addresses, at different scales, the planning and management of such a linear regional with the main purpose of establishing the location of new cruising ports. However, aside the conclusion that the location and development of passenger points is/should be directly linked with the proximity of cultural heritage and points of interests directed at tourism, no other conclusion is raised on the relationship between river cruise tourism and spatial planning at different scales and government levels. This is in my opinion a very poor conclusion and much more can be said and developed, namely in regard to the regional cooperation, the integration of different sectoral views and sector-wide policies, etc.

#4 Finally, I keep my feeling that the title of the article could be improved. It gives little insight on the background discussion and no clue on spatial planning is given.

#5 Please provide page number for quotations.

Reviewer 4 Report

The article has been substantially modified and improved.
The introduction has been deepened.

The literature review has been improved.
The discussion and conclusion sections have also been improved, which has ensured that the article is more critically rigorous, and of greater interest to research in the tourism sector.

The topic is very interesting, the study is innovative, and it will certainly be mentioned in future studies that aim to investigate the topic of cruise tourism in a river context, with itineraries equally valued in terms of tourist resources.

In the abstract, "decission" appears and "decision" should be written.

Author Response

Responses are it the attachment.
